# Improving policy design and epidemic response using integrated models of economic choice and disease dynamics with behavioral feedback

Hongru Du [1,2][☯]*, Matthew V. Zahn [3,4][☯]*, Sara L. Loo [5], Tijs W. Alleman[5], Shaun Truelove[5,6], Bryan Patenaude [5], Lauren M. Gardner[1,6], Nicholas Papageorge[3], Alison L. Hill [6,7,8]

1 Department of Civil and Systems Engineering, Johns Hopkins University, Baltimore, Maryland, United States of America, 2 Department of Systems and Information Engineering, University of Virginia, Charlottesville, Virginia, United States of America, 3 Department of Economics, Johns Hopkins University, Baltimore, Maryland, United States of America, 4 McDonough School of Business, Georgetown University, Washington, District of Columbia, United States of America, 5 Department of International Health, Johns Hopkins Bloomberg School of Public Health, Baltimore, Maryland, United States of America, 6 Department of Epidemiology, Johns Hopkins Bloomberg School of Public Health, Baltimore, Maryland, United States of America, 7 Department of Biomedical Engineering, Johns Hopkins University, Baltimore, Maryland, United States of America, 8 Department of Ecology and Evolutionary Biology, University of Toronto, Toronto, Ontario, Canada

☯ These authors contributed equally to this work.
* hongrudu@virginia.edu (HD); matthew.zahn@georgetown.edu, matthew.zahn@jhu.edu (MVZ)

**Data availability statement:** This work does not use any data. Codes are publicly accessible at https://github.com/HopkinsIDD/epi-econ.

## Abstract

Human behavior plays a crucial role in infectious disease transmission, yet traditional models often overlook or oversimplify this factor, limiting predictions of disease spread and the associated socioeconomic impacts. Here we introduce a feedback-informed epidemiological model that integrates human behavior with disease dynamics in a credible, tractable, and extendable manner. From economics, we incorporate a dynamic decision-making model where individuals assess the trade-off between disease risks and economic consequences, and then link this to a risk-stratified compartmental model of disease spread taken from epidemiology. In the unified framework, heterogeneous individuals make choices based on current and future payoffs, influencing their risk of infection and shaping population-level disease dynamics. As an example, we model disease-decision feedback during the early months of the COVID-19 pandemic, when the decision to participate in paid, in-person work was a major determinant of disease risk. Comparing the impacts of stylized policy options representing mandatory, incentivized/compensated, and voluntary work abstention, we find that accounting for disease-behavior feedback has a significant impact on the relative health and economic impacts of policies. Including two crucial dimensions of heterogeneity—health and economic vulnerability—the results highlight how inequities between risk groups can be exacerbated or alleviated by disease control measures. Importantly, we show that a policy of more stringent workplace testing can potentially slow virus spread and, surprisingly, increase labor supply since individuals otherwise inclined to remain at home to avoid

**Funding:** This work was funded by the US National Science Foundation (Award 2229996 - HD, MVZ, SLL, ST, BP, LMG, NP, ALH) and the Centers for Disease Control and Prevention (76NU38FT000012 - SLL, TWA, ST, LMG, ALH). The funders had no role in study design, data collection and analysis, decision to publish, or preparation of the manuscript.

**Competing interests:** The authors have declared that no competing interests exist.

infection perceive a safer workplace. In short, our framework permits the exploration of avenues whereby health and wealth need not always be at odds. This flexible and extendable modeling framework offers a powerful tool for understanding the interplay between human behavior and disease spread.

## Author summary

Models help researchers and policymakers predict how infections spread and compare control strategies. However, current models neglect how behavioral choices (like social distancing or vaccination) influences and reacts to disease spread. We present a new model combining ideas from epidemiology and economics to describe feedback between individual decisions, population health, and economic outcomes. Simulated individuals evaluate their future infection risk and weigh the costs/benefits of possible actions. Different health or economic vulnerabilities lead to distinct trade-offs and behaviors. We model the early stage of COVID-19 when people had to choose between going to work and risking infection or staying home and losing income. More generally, our model provides a flexible tool for policymakers to compare interventions to reduce disease, limit costs, and prevent disparities.

## Introduction

Infectious disease transmission is driven by human behavior, which brings people in contact with the pathogens we host. For most of human history, behavioral modifications to reduce transmission, such as quarantine and isolation, were the main methods of infection control [1]. Even today, the choice to get tested, vaccinated or take medication drives individual and collective risk for many diseases [2]. During the early years of the COVID-19 pandemic, policies to induce widespread behavior changes such as business/school closures, stay-at-home orders, and travel bans were common [3]. While these interventions dramatically reduced disease spread and healthcare burden for some time, they also caused substantial disruptions to well-being [4]. Thus, a recurring question facing policy-makers has been "How do we reduce disease burden while simultaneously mitigating the social and economic costs of doing so?"

Disease transmission models are powerful tools for informing control policy, as evidenced by their widespread use for infections such as COVID-19, HIV, influenza, measles, and malaria (e.g. [5–11]). Historically, these models track changes in the portion of a population at risk of infection and in different stages of disease progression (e.g., the Susceptible-Infectious-Recovered (SIR) model, [12]). Transitions between stages are determined by composite parameter values informed by epidemiological observations—such as the probability of transmission per time period for a given population density or the average duration of infectiousness—which obscures the specific impact of human behavior or the generative process governing it. During COVID-19 and to some extent before, data such as contact surveys [13,14], mobility metrics [15,16], or real-time vaccination tracking [17] allowed models to modify parameter values using these data-derived correlates of behavior, often at high temporal or spatial resolution [18] or stratified by known risk factors like age [19,20]. However, these approaches abstract from the mechanisms behind individual-level decision making and

thus fail to capture the dynamic trade-offs between health and other aspects of well-being that individuals face as disease burden and control strategy evolve.

To adequately capture health-wealth trade-offs, we need modeling frameworks that account for the complex interactions between disease propagation and the behaviors that drive it, including *feedback loops* (whereby behavior change leads to changes in disease dynamics that in turn lead to further shifts in behavior), *externalities* (whereby individual choices have an impact on others in society), and *heterogeneity* in decision making (whereby individuals may face different trade-offs depending on their health or economic vulnerability). Otherwise, it is difficult to generate reliable projections of disease transmission or to evaluate the welfare consequences—including economic costs—of prospective public health policies.

Health economists have long integrated infectious disease models in cost-benefit and cost-effectiveness analyses to guide public health policy [21,22]. However, these approaches often rely on simple models of disease spread that rarely consider feedback among disease prevalence, individual behavior, and public policy [23]. Projected health outcomes are typically converted to disability- or quality-adjusted life years [24], metrics which do not encompass overall well-being [25,26]. Furthermore, the economic analyses accompanying these studies tend to focus narrowly on direct medical costs and particular indirect costs such as lower productivity, neglecting costs stemming from behavioral changes, income loss, and the broader disutility of policy constraints. This omission can lead to underestimates of the full cost of the disease or policies to curb it, along with inaccurate predictions about behavior and thus disease spread.

The study of how individuals weigh trade-offs to make decisions in a variety of circumstances—including infectious disease outbreaks—is a substantial part of research in economics. Prior work has integrated disease dynamics into models of human behavior related to labor supply, consumption, and risky behaviors (see e.g., [27–32]). With a primary goal of better understanding human behavior, these studies have placed less emphasis on the epidemiological components, potentially leading to misspecifications of how diseases are contracted, transmitted, or progress. Nevertheless, there are notable examples where explicit behavioral modeling has been used to recover otherwise hidden health dynamics [33]. Despite capturing how behavior endogenously responds to prevailing disease conditions, these models thus tend to be ill-equipped to forecast disease dynamics, which in turn can lead to inaccurate forecasts of behavioral responses and evaluations of intervention policies.

To address these challenges, a growing body of research in behavioral epidemiology and economic epidemiology has begun developing integrated frameworks of disease spread and human behavior (e.g., [34–45]). Prior models have included reasonable approximations to both pathogen transmission and behavior. For example, traditional infectious disease models have been extended to include heuristic functions for changes in contact rates with disease burden (e.g., [46–49]), to model the spread or "imitation" of behaviors contemporaneously with infection (see e.g., [50,51]), or to consider behavior as a game-theory problem where disease levels are static on the timescale of decision making and large groups of the population collapse into a small number of "players" all making the same sets of decisions (e.g., [52–56]).

Inspired by the needs of policymakers during the COVID-19 pandemic, new approaches to model behavior and disease spread have emerged. One leverages macroeconomic models assuming non-infected individuals supply labor thereby contributing to aggregate output [57–64]. These frameworks forecast policy-relevant indicators such as unemployment and gross domestic product, but, without a formal model of individual-level decision making (or by assuming imitation), cannot fully capture the feedback and trade-offs that influence

economically-relevant behavior or adequately capture the welfare consequences of policy. Approaches that do directly model how individuals make decisions often employ fixed decision rules, sometimes informed by data, to predict how behavior will respond to prevailing disease conditions [65–70]. Such approaches are not designed to capture how individuals re-optimize under counterfactual policies or disease scenarios.

A handful of prior papers have incorporated formal models of behavior where decisions are made to optimize a measure of well-being or utility with potentially incomplete information, and can thus project how behavior endogenously responds to changing disease and policy conditions [71–74]. However, these studies have two main limitations. First, some have ignored individual heterogeneity in vulnerability to disease (e.g., preexisting conditions) or economic hardship (e.g., low income), instead differentiating individuals *only* by their infection state [71–73]. Capturing population heterogeneity is critical not only for quantifying the distributional benefits and burdens of different policy interventions, but also for accurately predicting population-level disease spread, as concentration of infection in risk groups promotes persistence despite control efforts. A second limitation of this body of work is the use of non-standard or inflexible approaches to describing infection spread [29,31,74–76]. For example, Brotherhood et al. [74] capture important margins of individual heterogeneity in their model of behavior, but make limiting assumptions in their epidemiological model (e.g., random mixing, no group stratification, calibrated disease dynamics).

In this paper, we present a dynamic feedback-informed epidemiological model (FIEM) that draws from economics and epidemiology to integrate infectious disease dynamics with individual behavior (Fig 1). Our framework classifies individuals based on their infection state variables, such as time-varying infection status (e.g., susceptible, infectious) and non-infection state variables, which include decision status (e.g., choice to work or engage in social distancing), as well as by a set of other state variables that may be fixed or time-dependent (e.g., demographics, health vulnerability, socioeconomic profile). The two core components of the dynamic mathematical model- the risk-stratified model of disease transmission and the individual-level model of decision-making- determine how individuals' infection and

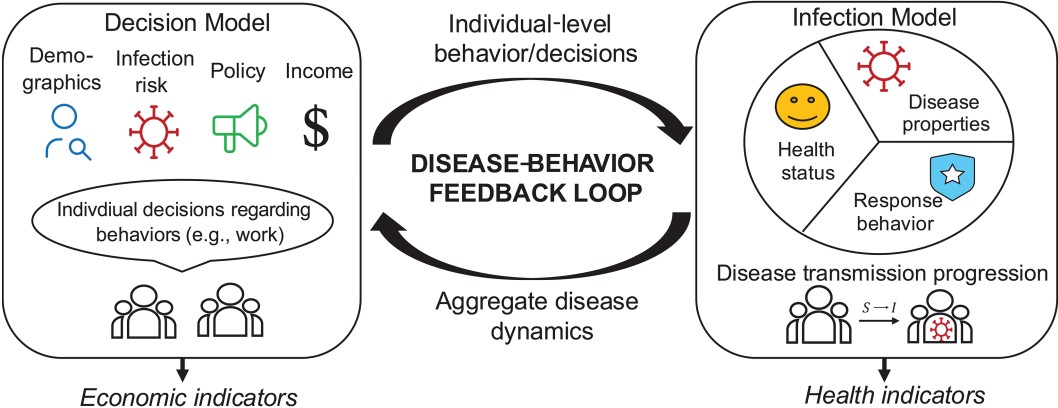

**Fig 1. Conceptual overview of the Feedback-Informed Epidemiological Model (FIEM).** The decision model simulates individual behaviors based on perceived disease dynamics, economic costs, policies, and demographics. These behaviors drive aggregate economic outputs and also disease dynamics. Concurrently, the infection model stratifies individuals into risk groups based on their decisions and individual factors, tracking disease transmission, progression, and recovery. The resulting disease dynamics again affect future individual behaviors and the course of the epidemic. A detailed visualization of each model component is presented in Fig 2.

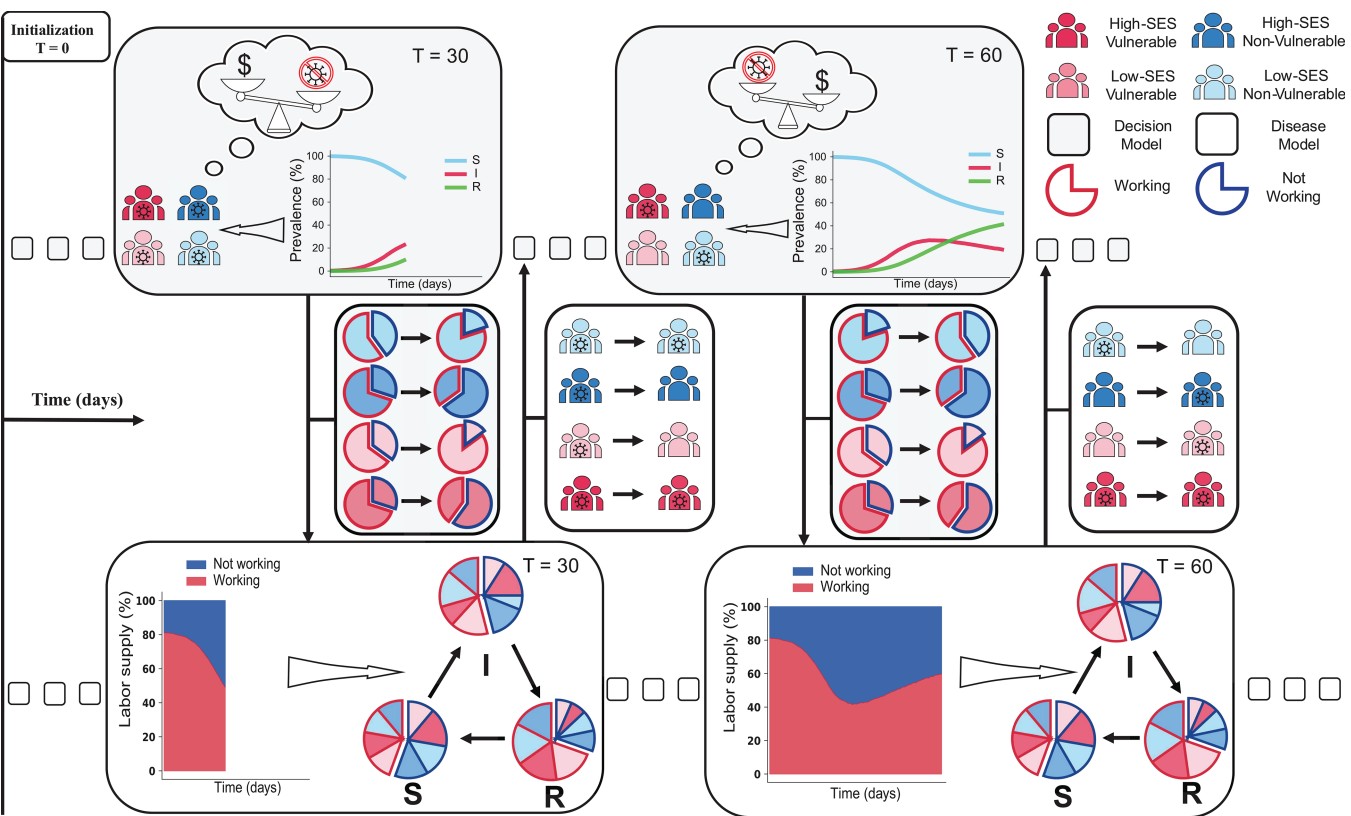

**Fig 2. Detailed framework of the Feedback-Informed Epidemiological Model (FIEM).** As an example scenario, we model the decision to work ('work choice') during an outbreak of an infectious disease transmitted between casual contacts. Individuals are stratified by their vulnerability to disease (red vs blue) and socio-economic status (SES) (bright vs pale color). Over time (left to right of figure) they repeatedly weigh the trade-off between income from working and risk of disease. Top row: Decision-making model. Each time period (grey hashed box), individuals make a decision (e.g., to work or abstain from working), based on their own infection status, population-level disease burden, and expected individual utility. Middle row: Decisions influence the distribution of individuals across risk groups (pie charts), which feeds into the epidemiological model (downward arrow). Bottom row: Epidemiological model. Risk groups membership, which depends on individual characteristics and decisions, influences the probability of transitioning between infection states (e.g., S - susceptible/uninfected, I - infected/infectious, R - recovered/immune). The updated individual and population-level disease burden then influences decisions made in future periods (upwards arrow and middle row). The framework is adaptable to other behaviors and population characteristics.

decision states evolve over time (Fig 2). We designed FIEM to be flexible, allowing the disease and decision models to be extended in many possible directions, such as adding more infection states (e.g., asymptomatic, mild-symptomatic), incorporating additional decision sets (e.g., compliance with mask mandates, willingness to take vaccine, allocating time between work and leisure), or specifying new state variables to further differentiate individuals. Together, these characteristics make FIEM a powerful and flexible tool for policy analysis; generating predictions about disease spread and economic consequences that capture endogenous individual decision making, and allowing for analysis of the impact of policy interventions across different types of individuals in the population. Because the model explicitly incorporates feedback between individual decisions and the aggregate spread of illness, it can generate counterintuitive results. For example, we find that a policy testing individuals who choose to work can reduce infection rates while increasing labor supply and income since individuals who might otherwise stay home perceive the workplace as safer. While this outcome depends on model assumptions, it challenges the common perception that health and economic goals are inherently at odds, and illustrates the utility of a unified framework for uncovering both direct and indirect effects of policy.

## Materials and methods

In this section we summarize the conceptual structure of the model and the application to the decision to work during the early phase of COVID-19. Details of our model including the motivation, mathematical formulations, and computational approaches are provided in the Supplementary Materials. Model code is available at https://github.com/HopkinsIDD/epi-econ.

### Components of the feedback-informed epidemiological model

**Individual-level decision model.** Individuals make decisions each period, such as whether to work or not, based on their perception of infection levels and the expectations they form about how their choices influence their future risks of getting infected (Fig 2). Individual decision-making is modeled as a discrete choice to maximize expected lifetime utility, a well-established method in economics [77–79] that aligns with other methods of modeling behavior from psychology and sociology [80]. Individuals make decisions dynamically—their actions are optimal from their individual perspective given how these choices influence the current period's utility (a function of their infection and non-infection state variables along with their choice) as well as the expected (since future outcomes are probabilistic) present discounted stream of future utility. An optimal decision thus reflects the utility payoffs, information, and beliefs structure of the model, which can be flexibly specified within this modeling framework. Individual and population state variables evolve each period based on the decisions made by individuals in the population.

**Risk-stratified infection model.** Each period, individuals in FIEM are classified into a discrete set of risk groups based on their behavioral choices and non-infection state variables (Fig 2). The risk groups are used to construct a stratified compartmental model of infection spread, which tracks at a minimum the proportion of each risk group that is susceptible or infected, but may also track symptom severity, degree of immunity to infection, or diagnostic status, for example. Parameters governing the transitions between disease states can vary by risk group (e.g., contact rates, susceptibility to infection or severe outcomes, duration of infectious period), and individuals may preferentially make contact and thus transmit to others in similar risk groups. The dynamic infection model simulates disease spread and progression to determine the distribution of infection states at the end of each period.

**Disease-decision feedback loop.** The core of our model lies in the dynamic feedback loop between individual behavior and the distribution of disease states in the population. Aggregated individual decisions in combination with baseline characteristics determine the distribution of people across risk groups, which affects individuals' risks of acquiring, transmitting, and developing severe outcomes as a result of infection. This subsequently alters overall disease dynamics, and the optimal individual behavior going forward. This cyclical process captures the complex interplay: the infection level in the population influences individual-level behavior, and those behavioral responses in turn reshape the trajectory of the disease in the population.

### Decision scenario, model equations, and parameterization

To demonstrate the capabilities of FIEM, we designed a simple scenario to capture one of the core trade-offs faced during the early stages of the COVID-19 pandemic: the decision to work and earn income or stay home and minimize disease risk (see Supplementary Methods for details).

Disease spread in the population is described by a risk-stratified 'SIRS' (susceptible, infectious, recovered, susceptible) model [81], where individuals begin as uninfected and susceptible to infection (S), and may become infected and infectious (I) after contact with another infected individual. Infected individuals eventually recover (R) and develop immunity to reinfection, which over time can wane leading them to return to a susceptible (S) state.

Each period, if an individual chooses to work, they earn income but are more likely to contact infectious individuals, become infected, and incur costs (monetary and otherwise) related to infection. We include strong health-wealth trade-offs by incorporating two additional margins of individual heterogeneity—socioeconomic status (SES, low or high) and vulnerability to the disease (vulnerable or non-vulnerable). The combination of an individual's socioeconomic status (SES), vulnerability to severe disease, and decision to work determines their risk group. The rate at which susceptible individuals in risk group $g$ become infected (the "force of infection", FOI) is:

$$\text{FOI}_g = \beta \sum_{g_2 \in G} \mathbb{C}_{g,g_2} I_{g_2}(t)/N. \tag{1}$$

Here $\beta$ is the probability of disease transmission per contact per time, $\mathbb{C}_{g,g_2}$ is the propensity for contact between individuals in risk group $g$ and those in risk group $g_2$, $I_{g_2}$ is the number of infected individuals in risk group $g_2$, and $N$ is the total population size. Contacts ($\mathbb{C}$) are higher among individuals who choose to work and for those with low SES, and there is a degree of preferential mixing within risk groups.

Risk group membership changes dynamically as individuals decide whether to alter their behavior (in this case, decision to work) in response to their assessment of the potential costs and benefits. In time period $t$ individual $m$ has utility $u$ (overall well-being including health and income/expenses) specified as

$$u(z_{mt}, d_{mt}) = \log c_{mt}(d_{mt}) - (d_{mt} + i_{mt} p_c)\theta_h h_{mt} + i_{mt}\theta_x(1 + VUL_m\theta_v). \tag{2}$$

In this function, the state vector $z$ includes individuals' infection status (e.g., susceptible), socioeconomic status (e.g., high-SES), and vulnerability status (e.g., non-vulnerable). Although the components of $z$ appear explicitly in the utility expression, we retain $z$ as a shorthand to represent the individual's full risk and health profile. The decision to work in this time period is tracked using the indicator variable $d$. Abstaining from work reduces income, which in turn reduces how much an individual can consume and thus utility from consumption ($c(d)$). Low SES individuals experience greater reductions in consumption when abstaining from work. $i$ is an indicator variable for whether they are currently infected, $\theta_x$ is the utility cost of infection, $VUL$ is an indicator variable for belonging to the high vulnerability risk group, and $\theta_v$ is the increase in the disutility of infection for vulnerable individuals. We assume there is a baseline hassle cost of working given by $\theta_h h$ (fixed and random effects), while $p_c$ describes the increase in this cost if infected. With this formulation, the costs of working for a susceptible individual (currently uninfected) are the increased probability of future income loss and the disutility of becoming infected. The costs for an infected individual that chooses to work are the ongoing costs associated with disease symptoms or the stigma associated with being infectious.

Each timestep, each individual solves a dynamic optimization problem to decide whether to change their behavior (engage in in-person work) [78]. The solution to this optimization problem - a probability distribution over decisions in the next time step - is given by the solution to a recursive Bellman equation for the value function $V$ (total utility over an infinite

time horizon, present-discounted at rate $\kappa < 1$ )

$$V(z_{mt}) = \max_{d_{mt} \in \mathcal{D}} \left\{ u(z_{mt}, d_{mt}) + \kappa \sum_{z_{mt'}} \mathbf{P}(z_{mt'} | z_{mt}, d_{mt}) V(z_{mt'}) \right\}. \tag{3}$$

The term $\mathbf{P}(z_{mt'} | z_{mt}, d_{mt})$ encodes the dynamic infection model, describing the probability that an individual ends up in the state $z_{mt'}$ conditional on being in state $z_{mt}$ and making the decision $d_{mt}$ (e.g., moving from infection state S to I, given an individual goes to work). With additional assumptions to simplify the form of the value function, the simulation is conducted by an iterative algorithm that alternates between solving the optimization problem and updating the disease trajectory.

As a proof of concept, FIEM has not been fully validated against real-world data; instead, its parameters are sourced from previous studies grounded in empirical observations. We use an infectious period of $\approx 7$ days, an average duration of immunity of $\approx 6$ months, and a basic reproduction number ($R_0$) of 2.6 (an effective average over risk groups at baseline levels of workforce participation). To parameterize the term of the utility function, we assume the average non-vulnerable individual would be willing to pay $\sim \$6,000$ per day to avoid infection (relative to a mean daily income of \$180), based on prior estimates of the value per statistical case (see Supplementary Materials, pp. 13-14). A vulnerable individual would be willing to pay triple this amount to avoid infection (all monetary values in the paper are expressed in U.S. dollars (USD)). If an individual with low-SES chooses not to work they would have to reduce their consumption by 85%, while a high-SES individual would forgo 75% in the same situation. A detailed formulation and explanation of the variables, equations, and parameter values is provided in the Supplementary Materials.

For simplicity, we don't explicitly model working from home but our parameterization indirectly incorporates its main effect: reducing work contacts is less costly for high-SES individuals. When making decisions, we assume that individuals can accurately assess their own infection status, as well as their short-term risk of infection conditional on both their decision to work and the population-prevalence of infection (which we assume is correct but delayed by a one week lag in case reporting). FIEM can easily accommodate alternative assumptions about the information available to individuals and their understanding or beliefs.

## Results

### Dynamic behavior modification alters epidemic trajectory

Awareness of disease transmission in the community triggers individuals to make decisions to reduce the costs of being infected, and our integrated epidemic-behavior model (FIEM) captures this dynamic feedback endogenously (Fig 3). Compared to a traditional "fixed decision" epidemic model where the proportion of the population working is constant (i.e., constant contact patterns) for the duration of the outbreak, under FIEM, workforce participation drops quickly after the outbreak starts, resulting in slower initial epidemic growth and a lower peak (i.e, behavioral feedback naturally "flattens the curve"). Early on, knowingly-infected individuals choosing not to work due to the additional costs of working while infected are the main driver of the reduction in epidemic growth rate, but as infection prevalence increases, susceptible individuals avoid work due to the perceived risk of infection. In both cases, the drop in workforce participation results in fewer contacts between susceptible and infectious individuals and thus fewer new infections. Longer-term, in the absence of additional interventions, the proportion working is predicted to increase again as the peak recedes, but infection persists leading to a lower working population than before the outbreak. When both the standard

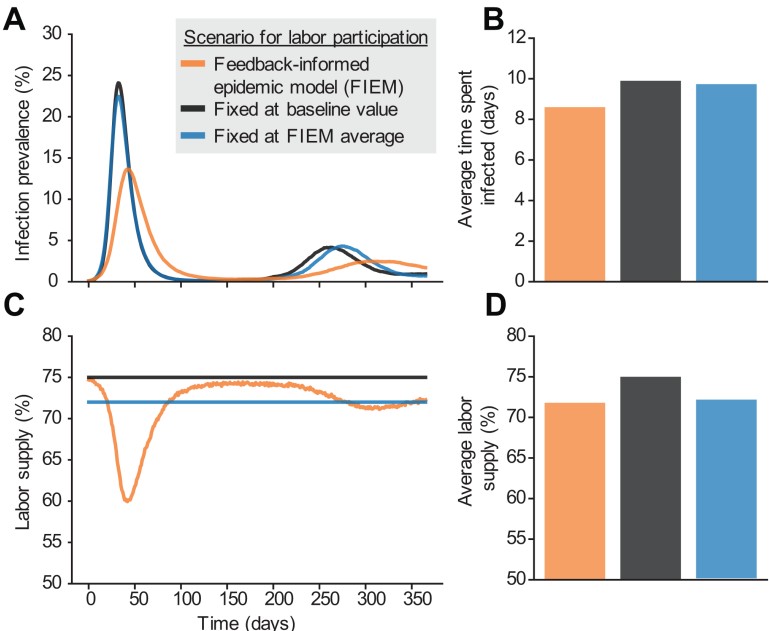

**Fig 3. Effect of endogenous behavior change on disease dynamics.** The time-course and time-average of infection levels (A,B) and workforce participation (C,D) for three model scenarios. In the feedback-informed epidemiological model ('FIEM', orange), individuals dynamically decide to work or abstain from work based on perceived costs and benefits, altering the degree of workplace transmission and thus population-level infection burden (initial working 75%, minimum working 60%, average working 72%, $R_0 \sim 2.05$, peak prevalence 13.7%). Two alternative fixed decision models are included for comparison: one in which the proportion of the population working each period is held constant at the pre-outbreak level (black, 75% working, peak prevalence 24.1%, $R_0 \sim 2.62$), and another where the work level is held constant at the average value observed in the feedback informed model after 1 year (blue; $R_0 \sim 2.52$, peak prevalence 22.5%, 72%, blue). $R_0$ values are estimated by fitting the logarithmic infection curves for the first 20 days. Note that panel B shows the cumulative average days each individual spends in the infectious state over a one-year SIRS simulation (allowing reinfections), so values can exceed the 7-day mean duration of a single infection event.

and feedback-informed models are parameterized to give the same average proportion of the population working over a year-long simulation period, FIEM predicts fewer infections. This simple comparison shows how including endogenous behavior can alter predictions of disease trajectories.

The predicted impact of behavior on disease dynamics depends on the underlying assumptions of the model, in particular, the health and wealth "payoffs" individuals weigh in their decision-making process (Fig 4). Infections that transmit more efficiently cause earlier and higher peaks and trigger earlier and more dramatic reductions in the number of susceptible individuals choosing to work (Fig 4A). If contacts at work are a larger portion of total contacts, meaning the majority of potential exposure to infected individuals occurs at work, a greater proportion of individuals choose not to work. However, within the predefined sensitivity range, the resulting epidemic curve shows no substantial deviation from scenarios where work contacts are less prevalent (Fig 4B).

The "utility cost of infection" captures the value per statistical case of COVID-19. This cost is intended to account for the possible clinical outcomes of an infection, reflecting the experience of typical symptoms as well as rare but costly severe outcomes. While this set-up limits the impact of infection to arise through the utility function and not other channels such as reduced consumption because an individual needs to seek out and pay for medical care,

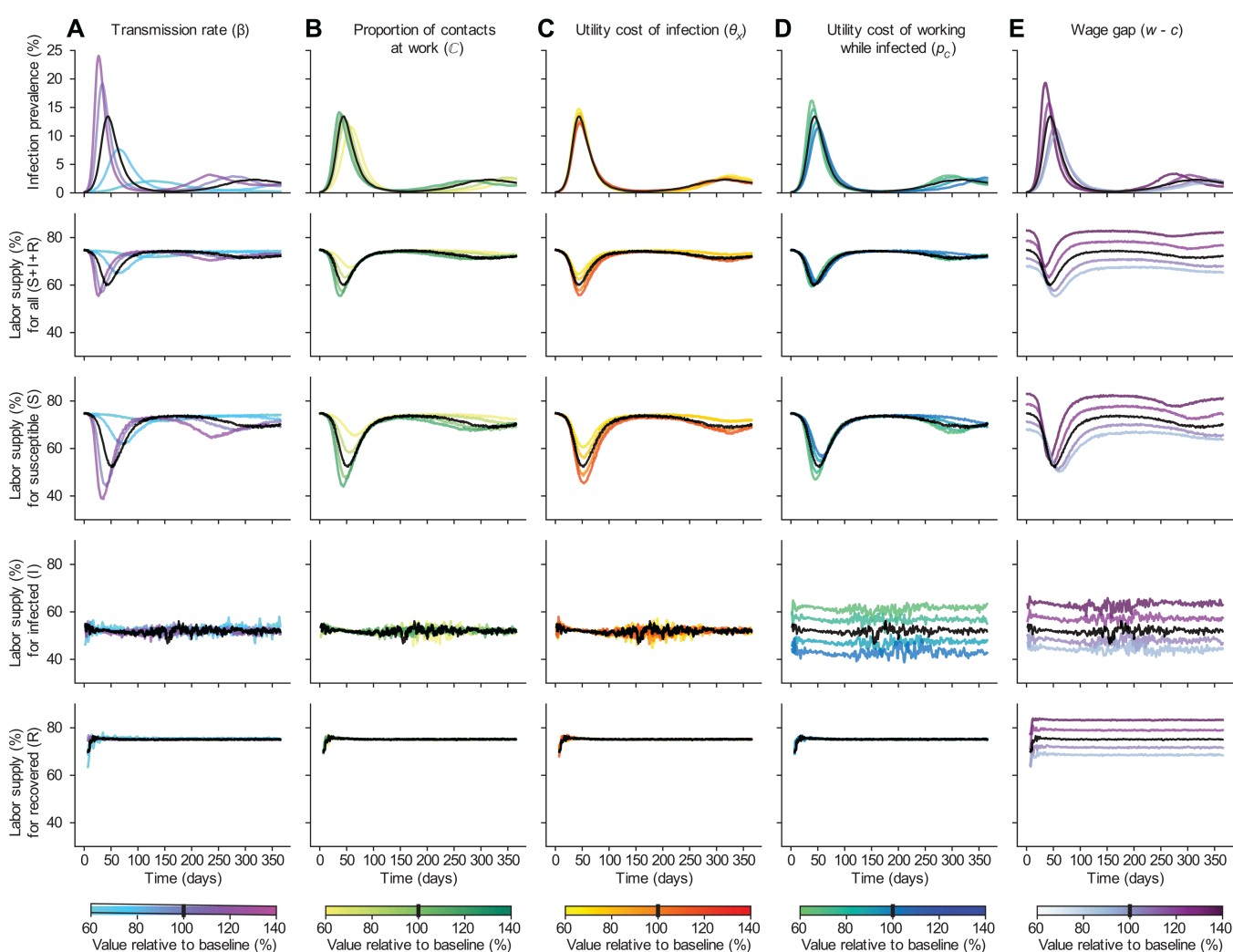

**Fig 4. Mechanisms driving individual behavior and disease feedback.** The time course of infection levels (first row) and workforce participation (for the entire population - second row - or stratified by infection status - third to fifth row) for model scenarios varying a single parameter value while others are fixed. Black curves in each panel show row outcomes using the baseline parameters (Fig 3, Tables A–B in S1 Text). Parameters are varied by column: A) transmission rate (and thus $R_0$), B) proportion of all contacts occurring at work (holding $R_0$ constant), C) utility cost of infection, D) utility cost of choosing to work while infected, E) the wage gap (the difference in consumption when working vs not working). For all parameters, ranges extend from 40% below to 40% above baseline value.

it preserves our ability to assess the core health-wealth tradeoff that motivates our scenario. As this cost increases, susceptible individuals have stronger incentives to avoid the increased risk of infection at work. Thus, the number of individuals working drops lower once infection becomes common and the epidemic peak is blunted (Fig 4C). In contrast, the "utility cost of working while infected" represents the additional cost of working for infected individuals. As this cost increases, infected individuals are more likely to stay away from work, reducing their contact with others. More infected individuals choosing to stay home leads to a decline in the early epidemic growth rate as well as the peak infection rate (Fig 4D). Importantly, higher values of the "utility cost of working while infected" lead to an *increased* labor supply of susceptible individuals; infected individuals optimally choose to abstain from work; and the risk of acquiring infection at work thus decreases. Finally, "wage loss" (i.e., the difference between

individuals' income if they do or do not work) further influences work choice decisions. Thus, greater wage losses create a stronger incentive to work despite illness or risk of infection, since consumption (a component of the utility function) increases with income from wages. As employment increases, so does disease transmission, causing larger epidemic peaks (Fig 4E). This pattern is partially driven by the design of this model scenario, which abstracts from financial savings, but we note that this incentive would persist in a model that allowed individuals to reduce the variation in their consumption each period by relying on their savings. The impact of other parameters—such the time lag in individuals' information on population-level disease burden and additional utility cost of infection for vulnerable individuals—are shown in Figs E and F in S1 Text.

### Consequences of heterogeneous health-wealth trade-offs

To demonstrate our framework is capable of accounting for the inherent heterogeneity of real-world populations, we next examine how variation among individuals in vulnerability to disease and socioeconomic status impacts behavior, shapes trade-offs, and subsequently influences the epidemic trajectory. In our scenario, high-SES individuals make more money if working and have a lower opportunity cost of not working, which we attribute to omitted factors, such as savings or having jobs that allow work-from-home arrangements. Low-SES individuals have more contacts at work and preferentially contact other low-SES individuals. Vulnerable individuals face higher utility costs from infection (i.e., have a higher likelihood of progression to more severe infection), but have no difference in per-exposure susceptibility to acquiring disease. We evaluated the infection trajectory predicted by our feedback-informed model for a baseline population with an even distribution of individuals across four risk groups (i.e., non-vulnerable/high-SES, vulnerable/high-SES, non-vulnerable/low-SES, vulnerable/low-SES) (Fig 5, see Fig H in S1 Text for alternative distributions yielding similar results).

We start by analyzing individual incentives to preserve their economic well-being. Low-SES individuals face a stark and disproportionate trade-off between economic needs and

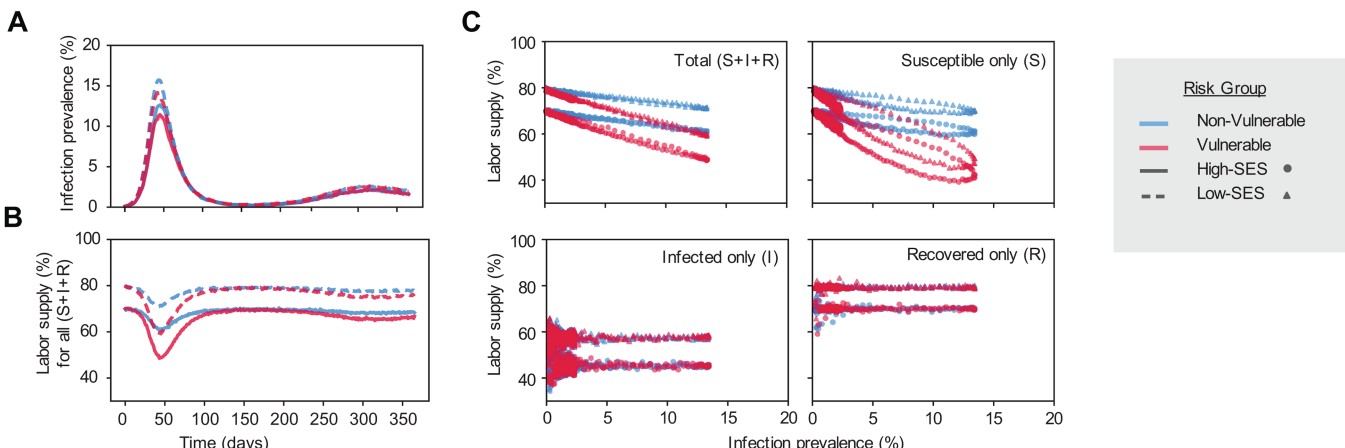

**Fig 5. Disease-decision dynamics across heterogeneous risk groups.** The dynamics of infection levels and the decision to work in a population stratified by vulnerability to severe disease if infected (vulnerable - red, non vulnerable - blue) and socioeconomic status (high SES - solid line or circle, low SES - dashed line or triangles). A) Infection prevalence over time; B) Fraction working over time; C) Share working vs infection prevalence for individuals of all infection statuses (entire population), susceptible individuals only (S), infected individuals only (I), and recovered/immune (R) individuals only. In this scenario, the four risk groups are of equal size.

health preservation. Low-SES individuals choose to work during the early outbreak stage despite infection risk, driven by their urgent need to meet necessities. As a consequence, low-SES individuals experience higher early exponential growth rates and epidemic peaks (Fig 5, dashed curves and triangles). Conversely, high-SES individuals exhibit more cautious behavior, with more individuals abstaining from work for a given infection level, reflecting their financial ability to prioritize health over wealth (Fig 5, solid curves and circles). These results underscore the need to consider socioeconomic inequalities when designing public health policies.

Next we analyze individuals' incentives to protect their health. Vulnerable populations exhibit stronger self-protective behavior due to the higher risks associated with infection. This feature creates an incentive for these individuals to not work, forgoing some consumption, during the period of highest disease burden (Fig 5, red curves and shapes). Since we assume individuals have perfect information about their current health state, only the susceptible group responds to infection risks (Fig 5C). This assumption can be relaxed to capture settings where infected individuals may be unaware of their status and either i) abstain from work believing it could prevent infection, and ii) continue to work without experiencing any of the utility costs of working while infected.

Our results highlight how key differences in health-wealth trade-offs experienced by different risk groups influence the joint trajectory of infection and behavior, as well as distributional consequences of the epidemic.

## Evaluating the impact of policy interventions

The goal of our framework is to provide a tool for analyzing disease control policies that incorporate endogenous behavior changes to improve prediction of infection burden, understand the distributional consequences of policies, calculate welfare, and identify optimal policies based on the specific needs and values of decision makers. To illustrate this potential, we encoded and compared four different policy interventions within our model (see Table 1 and Supplementary Materials): *labor restriction*, in which a portion of the population is constrained to abstain from work; *unconditional cash transfer*, in which a daily subsidy payment is provided to all individuals; *conditional cash transfer*, in which a daily subsidy payment is provided exclusively to individuals who choose not to work, and; *paid sick leave*, in which infected individuals who choose to abstain from work receive baseline wages and additional subsidy payments. We assume perfect compliance with policy recommendations and perfect knowledge of infection status.

We compare peak infection and employment outcomes under varying degrees of intervention for each policy (Fig 6). Labor restrictions have the largest marginal impact in reducing total and peak infections. The highest level of restriction we simulate (70%) lowers peak prevalence from 13.7% in the no-intervention scenario to 2.1% (Table 1, Fig 6A), but carries large economic burdens; translating to a $59 average loss of income per day per capita relative to a scenario where the disease outbreak occurs with no infection control policies in place and endogenous behavior change. Unconditional and conditional cash transfer policies also demonstrate considerable reductions in peak infection rates; at the highest payment levels (50% of the average wage), peak prevalence is reduced to 4.8% for unconditional and 2.7% for conditional transfers (Fig 6B–6C). These policies result in higher employment rates than labor restrictions, with unconditional transfers maintaining higher participation than conditional transfers. However, cash transfers have higher direct costs to the government than labor restrictions. Paid sick leave tends to have less impact on reducing peak and average infections than cash transfers, but it increases the average share of the population choosing to work

**Table 1. Summary of policy scenarios and resulting health and economic outcomes.** Metrics in parentheses are relative to the no intervention scenario. Each policy starts 20 days after the first infection (infection prevalence ~2.5%) and continues for 4 months before being relaxed. Total daily cost per capita includes both lost wages due to the disease (compared to a disease-free scenario with 75% working) and the cost of any subsidy payments provided. A detailed breakdown of costs, stratified by wage loss and subsidy payment, is in Table D in S1 Text. Each subsidy payment amount was benchmarked against the average wage (AW) in the population ($180 per day). All the values are calculated over the policy implementation period (day 20 to day 140).

| Policy | Description | Policy Scenario | Peak infection prevalence | Average infection prevalence | Average share working | Total daily cost per capita |
|---|---|---|---|---|---|---|
| No intervention | The simulation results were generated from the FIEM without applying any interventions. | No policy applied | 13.7% (1) | 8.2% (1) | 66.7% (1) | $13.0 (1) |
| Labor restriction | This policy randomly constrains a defined share of the population to remain at home, while the rest are free to choose whether or not to work. | 30% Labor restriction | 6.9% (0.50) | 5.2% (0.63) | 48.9% (0.73) | $36.3 (2.79) |
| | | 40% Labor restriction | 4.9% (0.36) | 3.9% (0.48) | 43.0% (0.64) | $44.4 (3.42) |
| | | 50% Labor restriction | 3.6% (0.26) | 3.1% (0.38) | 36.5% (0.55) | $53.2 (4.10) |
| | | 60% Labor restriction | 2.5% (0.18) | 2.3% (0.28) | 29.9% (0.45) | $62.1 (4.78) |
| | | 70% Labor restriction | 2.1% (0.15) | 1.8% (0.22) | 23.0% (0.34) | $72.0 (5.54) |
| Unconditional cash transfer | This policy offers a daily subsidy payment to all individuals regardless of choices, mirroring payments the American government provided during the COVID-19 pandemic. | Payment = 10% AW | 9.0% (0.66) | 6.5% (0.79) | 59.5% (0.89) | $37.9 (2.92) |
| | | Payment = 20% AW | 7.2% (0.53) | 5.6% (0.68) | 55.4% (0.83) | $60.5 (4.65) |
| | | Payment = 30% AW | 6.0% (0.44) | 4.8% (0.59) | 53.2% (0.80) | $81.2 (6.25) |
| | | Payment = 40% AW | 5.5% (0.40) | 4.5% (0.55) | 51.0% (0.76) | $101.8 (7.83) |
| | | Payment = 50% AW | 4.8% (0.35) | 3.9% (0.48) | 50.0% (0.75) | $121.3 (9.33) |
| Conditional cash transfer | This policy provides a daily subsidy exclusively to individuals who choose not to work, aiming to mitigate the economic consequences of that decision. | Payment = 10% AW | 7.6% (0.55) | 5.8% (0.71) | 57.5% (0.86) | $30.0 (2.31) |
| | | Payment = 20% AW | 5.2% (0.38) | 4.2% (0.51) | 51.8% (0.78) | $46.3 (3.33) |
| | | Payment = 30% AW | 4.0% (0.29) | 3.6% (0.44) | 47.3% (0.71) | $63.1 (4.85) |
| | | Payment = 40% AW | 3.7% (0.27) | 3.3% (0.40) | 43.4% (0.65) | $80.6 (6.20) |
| | | Payment = 50% AW | 2.7% (0.20) | 2.5% (0.30) | 40.8% (0.61) | $96.6 (7.41) |
| Paid sick leave | This intervention provides direct financial support to infected individuals who choose to stay home from work, thus directly targeting the health-wealth trade-off infected individuals face. | Payment = 10% AW | 10.4% (0.76) | 7.3% (0.89) | 67.3% (1.01) | $12.2 (0.94) |
| | | Payment = 20% AW | 8.6% (0.63) | 6.4% (0.78) | 68.1% (1.02) | $11.8 (0.91) |
| | | Payment = 30% AW | 7.4% (0.54) | 5.8% (0.71) | 68.8% (1.03) | $11.5 (0.88) |
| | | Payment = 40% AW | 6.7% (0.49) | 5.3% (0.65) | 69.3% (1.04) | $11.4 (0.87) |
| | | Payment = 50% AW | 5.9% (0.43) | 4.9% (0.60) | 70.0% (1.05) | $11.3 (0.87) |

(≈ 70%, vs 50–60% for the unconditional transfer and 40–58% for the conditional transfer).

We also evaluate the cost-effectiveness of each policy. Costs are defined as the net of subsidy payments and wage losses due to reduced labor supply, and expressed both as a dollar value and percentage relative to the no intervention scenario (Table 1, Figs I and J in S1 Text). We did not include other potential costs associated with these policies, such as the costs of administration, diagnostic tests, or enforcing restrictions. We evaluate effectiveness in terms of the peak infection prevalence, but other epidemiological metrics could also be used. For labor restriction and cash transfers, stronger versions of the policies which incur higher costs are associated with lower peak infection levels. Labor restrictions achieve equivalent peak infection reductions for lower costs than other policies (Fig 6E–6F). For example, under the example parameters used for this simulation, a labor restriction policy costing around $40 per person per day reduced peak infections to a third of the no intervention scenario, whereas achieving similar reductions costs close to $120 with an unconditional cash transfer. However, the paid sick leave policy deviates from this pattern, and uniquely achieves reduced infection rates and lower total costs as subsidy payments increase. For example, providing 50% of the average wage for a paid sick leave policy reduces the total daily cost per capita to $19 (0.76 of the cost with no intervention) and the average infection rate to 5.9% (0.43 of the peak size of no intervention). Paid sick leave accomplishes this infection reduction while also *increasing* the average amount of labor supply in the population, thereby reducing wage

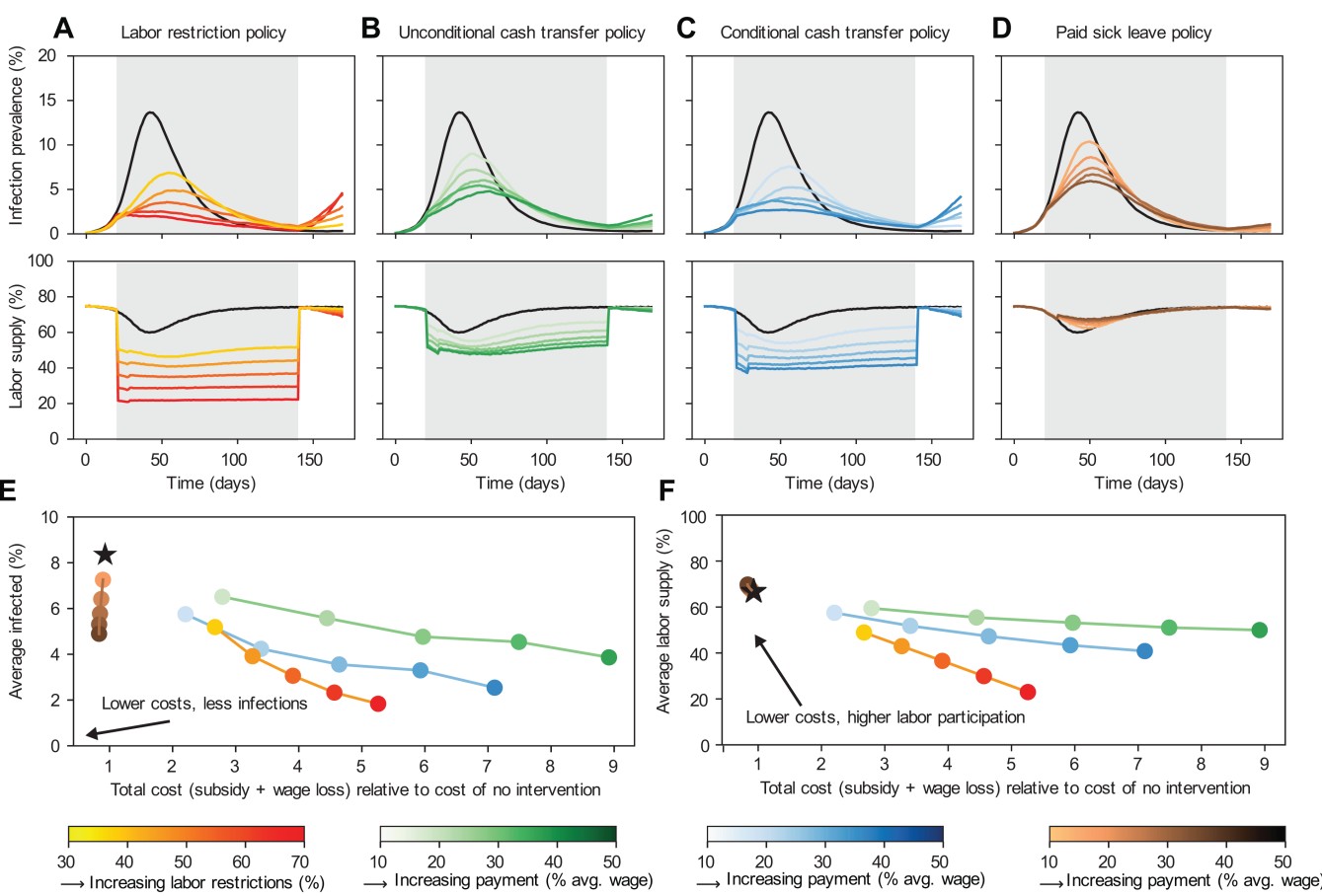

**Fig 6. Effects of policy interventions on population health outcomes and work decisions.** A-D) Population infection prevalence and share of the population choosing to work over time, for different policies. The black curves and stars are the baseline model with no intervention—no labor restriction and no additional payment. A) Labor restriction policy (yellow to red) that limits how much of the population is able to choose to work. B) Unconditional cash transfer (green) that is delivered to all individuals each period. C) Conditional cash transfer (blue) that is delivered to individuals who choose not to work each period. D) Paid sick leave policy (brown), which allows infected individuals to earn their full wage if they choose to not work while infected. The simulations start with no policy intervention in place. The intervention begins on day 20 and remains in place for 4 months (until day 140). E) Average share of the population infected while policy is in place versus cost of policy. F) Average share of the population choosing to work while policy is in place, vs cost of policy. Policy costs include the sum of government spending to fund transfers plus lost wages to individuals relative to their baseline wage earnings predicted by the model when there is no disease present.

loss costs. By giving infected individuals a strong incentive to not work, the risk of infection for a susceptible person declines and allows them to endogenously decide to work, an example of a positive externality of the policy. In reality, the cost-effectiveness of paid sick leave is complicated by the issue of accurate detection of infectious individuals and malingering. However, our framework's ability to model individual decision-making allows us to capture the core effects of this policy and could be expanded to include more details, providing valuable insights for policymakers who must consider both the intended and unintended consequences of their interventions.

We also use our framework to evaluate which policy designs are optimal for achieving pre-specified objectives (Fig 7). To do so, we construct a social welfare function, which specifies how to weigh the cost of the policy versus the benefit of fewer total person-days of infection and whether to impose a budget constraint for the policy's costs. These components may

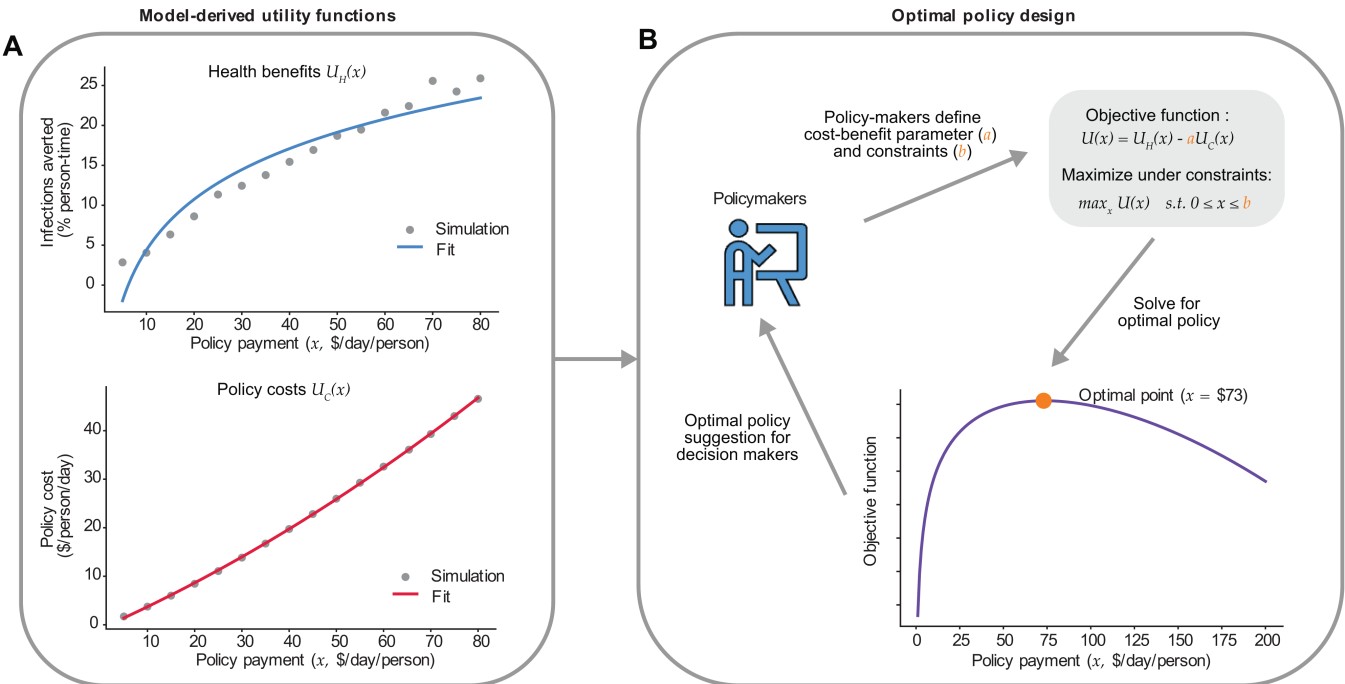

**Fig 7. Decision-maker informed optimal policy design.** Schematic diagram outlining how policymakers can use the feedback-informed epidemiological model to identify an optimal policy design. We use the unconditional cash transfer policy as an example, assessing its impact over a 4-month implementation period. A) The simulated health benefits $U_H(x)$ and policy costs $U_C(x)$ as functions of daily subsidy payment, fitted to generate continuous curves. Health benefits $U_H(x)$ are defined as the reduction in the average fraction of days an individual spends infected during the evaluation period. This is calculated as $(\sum_t I_t - \sum_t I_t^p)/(NT)$, where $I_t$ is the number of infected individuals on the day $t$ with no intervention, $I_t^p$ $N$ is the number of infected individuals on the day $t$ with implemented policy, $N$ is the population size, and $T$ is the duration of the evaluation period. The policy cost $U_C(x)$ is defined as the payment cost per capita per day over the evaluation period, given by $(\sum_t C_t)/(NT)$, where $C_t$ denotes the policy cost on day $t$. Increasing policy payment increases both health outcomes and the associated cost of the policy, creating a trade-off between the two. B) Policymakers decide how to numerically weigh the relative benefits and costs of each policy, and specify any monetary or political economy constraints. This allows for the definition of a single objective function $U(x) = U_H(x) - aU_C(x)$ that can be maximized to determine optimal payment amount, subject to the assumed parameters and defined constraints. In this example, we estimated the daily cost of infection per capita to be ~\$6,000 (see Supplementary Materials, Parameters section for a detailed calculation) so that a unit increase in $U_C(x)$ (\$1) would be equivalent to 0.00017 ($a = \frac{1}{5912} \approx 0.00017$) of a unit in $U_H(x)$. Increasing $a$ places greater emphasis on minimizing policy cost, while decreasing it prioritizes reducing infections averted. Note that the estimated values above are dependent on the length of the evaluation period. In this example, the optimal policy would be identified as a \$73 per person per day cash transfer, which would be expected to avert approximately 22 days of infection per 1,000 individuals in the population.

vary across scenarios or across policymakers. Once the social welfare function is specified, we can solve for the policy stringency or payment level that maximizes this function subject to its constraints. We demonstrate how to perform this type of analysis with the conditional cash transfer policy. We use a weight that reflects a willingness to pay \$1 per capita per day to reduce the total number of people infected per day by 2.71, which is based on the value per statistical case of COVID-19 used by the US Department of Health and Human Services [82]. Given these conditions, we find the optimal policy is a \$65 payment per individual per day. Defining the social welfare function for optimal policy is a complex decision, but FIEM can flexibly capture different weights or budget constraints policymakers must contend with when analyzing and designing policy.

## Distributional consequences of policies in heterogeneous populations

To evaluate the differential impact of policy interventions in subgroups experiencing different health-wealth trade-offs, we assess the impacts of each policy by socioeconomic status

and vulnerability to disease (Fig 8). Consistent with population level outcomes, we found that all policies effectively reduced infection levels in all groups, but failed to eliminate disparities in infection burden by SES status, although differences between the groups were slightly reduced for more stringent policies. However, we observed heterogeneous behavioral responses. Subsidy-based interventions disproportionately influence the behavior of low-SES groups, moving from the no-intervention scenario where they are more likely to maintain high labor supply despite infection risk to creating opportunities for them to acknowledge their higher infection risk at work and abstain. For example, conditional cash transfers cause a sharper decline in work participation for the non-vulnerable, low-SES group (from 59% to 35%) compared to the high-SES group (60% to 48%). Interestingly, the vulnerable, high-SES group exhibits modest reductions in choosing to work. This pattern is partially driven by the reduced labor supply of low-SES individuals, which makes the probability of infection lower

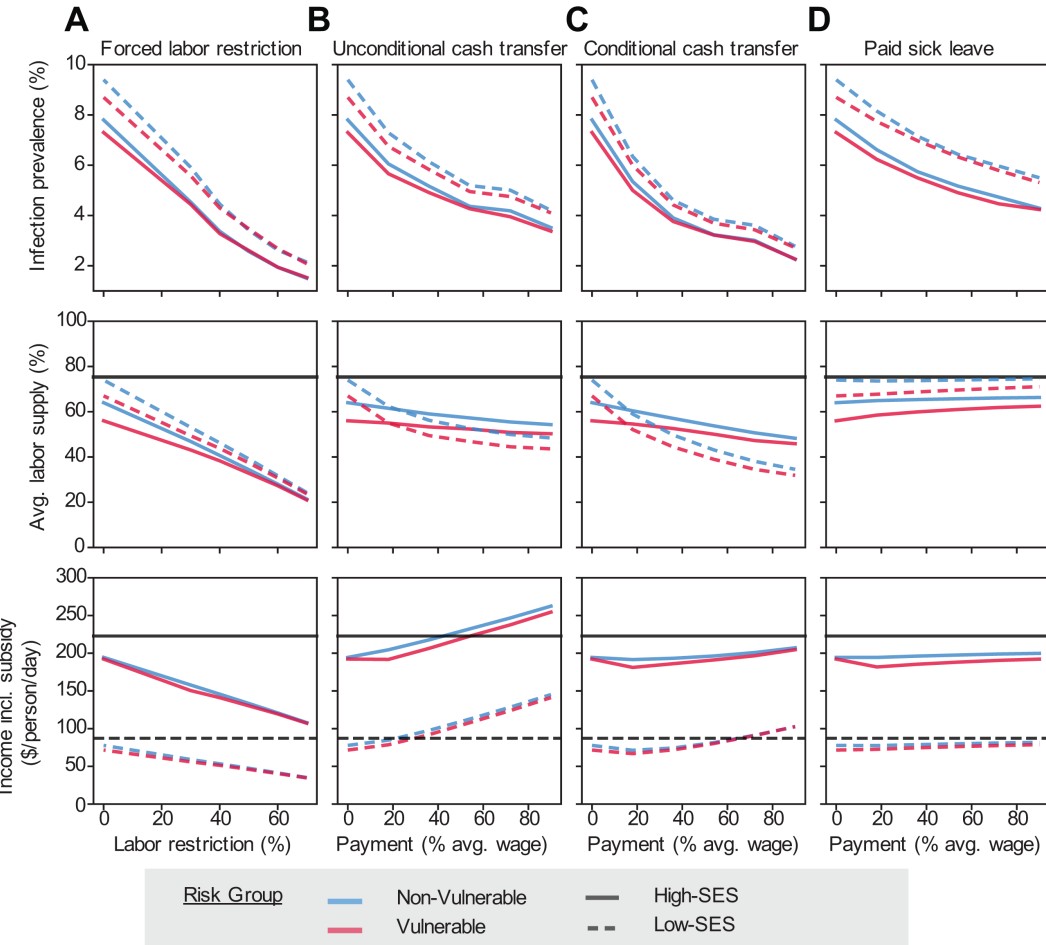

**Fig 8. Heterogeneity of policy impacts by risk groups.** Top row: Average percent of the population infected. Middle row: Average fraction of the population working. Bottom row: Income per capita per day, including wages and subsidies. A) Labor restriction policy limiting how much of the population is able to choose to work. B) Unconditional cash transfer delivered to all individuals each period. C) Conditional cash transfer delivered to individuals that choose not to work each period. D) Paid sick leave allowing infected individuals to earn their full wage if they choose not to work while infected. Heterogeneity in outcomes across risk groups is denoted by different curves: non-vulnerable (blue), vulnerable (red), low-SES (dashed lines), and high-SES (solid lines). The black lines give baseline income without disease.

for a high-SES individual if they opted to work. As the paid sick leave subsidy increased, labor supply is maintained despite reductions in infection, and this is largely driven by vulnerable populations, consistent with a reduction in their health-wealth trade-offs.

We also compare total income under each policy intervention relative to baseline income by population subgroups. Each policy generates notable differences. For example, low-SES individuals benefit substantially from the unconditional cash transfer policy relative to high-SES people. This pattern is driven by the size of the transfer relative to their labor earnings. Conversely, conditional cash transfer offers slightly higher benefits to high-SES individuals when payments are low, as they already have less incentive to work. Paid sick leave provides marginal but similar benefit to income for all groups, by increasing the share of the susceptible population of each group that opts to work. In contrast, labor restrictions significantly disrupt labor supply and negatively impact the income of individuals in all risk groups, with larger drops among low-SES individuals. This highlights how policies that appear effective by population-average metrics, like overall infection prevalence reduced per dollar of total cost, may exacerbate existing disparities and thus be considered suboptimal from an equity standpoint.

## Discussion

During a public health crisis, policymakers must balance population health and economic well-being. This task is complicated by subpopulations facing economic precarity or health vulnerability (or both), so any policy will have unequal distributional consequences. Useful models should predict the impacts of policy so that at minimum, we avoid inefficient policies, e.g., those with higher economic costs for no additional health benefits, or those with higher health burden but no economic benefits. In this paper we present a flexible modeling framework that captures the feedback between individual decision-making and infectious disease spread. We integrate a mechanistic model of disease dynamics (consistent with established best-practices in infectious disease epidemiology) with a formal model of individual decision-making based on forward-looking utility maximization (commonly used in economics). Our "feedback-informed epidemiological model" (FIEM) can flexibly encode the processes by which an individual's perceived risk of infection, among other factors, influences their behavior, which in turn impacts future disease propagation.

To illustrate the capabilities of FIEM, we designed a simplified scenario inspired by the early stages of the COVID-19 pandemic. Individuals decide whether to work or not based on the trade-off between health and economic well-being—abstaining from work lowers income but also reduces infection risk. Our model endogenously propagates the implications of this decision process, leading to slower epidemic growth and reduced peak disease burden but failing to achieve control. In the real world, both health vulnerability and economic precarity vary between individuals and affects incentives and behavior. When we expanded our scenario to include this heterogeneity, we predicted behavioral responses consistent with differential trade-offs, for example finding that individuals with lower socioeconomic status that are less vulnerable to disease maintain the highest rates of continued in-person work during an outbreak and experience the highest peak infection levels. This example scenario ignores many features of COVID-19 driven decisions, for example the high proportion of asymptomatic infections, limited testing, misinformation, the role of accumulated savings, the inability of many workers to leave and re-enter the labor force at will, and the purely pro-social motivations of some individuals for engaging in costly disease-avoidance behavior. However, even in a simplified model, our analyses underscore how individualized

health-wealth trade-offs and dynamic decision-making contribute to behavior change at the individual level, which carries direct implications for aggregate disease spread.

We used our model to evaluate the effects of four policies aimed at reducing disease spread: labor restrictions, unconditional cash transfers, conditional cash transfers, and paid sick leave. Each policy reduced peak and total infections, albeit through different mechanisms and at different costs to individuals and the government. Our key finding is that targeted policies incentivizing infected individuals to abstain from work can lower infection, as expected, but may also increase total labor supply by reducing the infection risk of working, thereby encouraging susceptible individuals to work. Finally, we demonstrate how this framework can identify optimal policy designs that balance the benefits of reduced infections with the associated policy costs, and decompose the heterogeneous impacts these policies have across different sub-populations. We believe this framework has the potential to advance policy-relevant disease modeling in multiple ways, including i) simultaneously outputting epidemiological, microeconomic, and macroeconomic metrics, ii) incorporating the impact of risk-avoidance behaviors that occur independently of mandated behavior change, and iii) centering considerations of equity in policy projections, by producing sub-group specific impacts.

The structure of FIEM allows it to be extended in many directions. Currently, our behavioral model makes two key assumptions: First, individuals engage in dynamic utility optimization with a low discount rate, thereby excluding behaviors such as impatience, hyperbolic discounting, and alternatives to dynamic optimization. Second, we assume individuals possess rational expectations (i.e., can accurately assess their current health status and future risks). However, in reality more complex decision models may be at play along with uncertainty over information about the outbreak. These assumptions can be tested with data on human behavior, beliefs, or information transitions, and FIEM can be readily adapted to incorporate these features as warranted. An additional assumption is the exclusion of strategic interactions among individuals. While it is possible to relax this feature, doing so may lead to multiple equilibria in the behavioral model, introducing methodological and computational complexities. Our model currently only considers individual decisions, whereas businesses and other institutions (e.g., schools) also engage in risk-avoidance decision making in response to disease. More critically, our current framework is not designed to describe macroeconomic processes that may also feed back with individual decision making during public health crises, such as changes in labor demand, economic growth or recession, inflationary processes, interest rate changes, among others. However, FIEM can incorporate macroeconomic models and produce integrated forecasts.

Beyond rational expectations and economic trade-offs, FIEM can be extended to incorporate psychological and emotional dimensions of decision-making, capturing a broader sense of well-being than just health or financial. For example, emotional costs such as social isolation, fear, or stigma can be modeled as additional non-monetary terms in the utility function. Individuals who isolate at home may experience a utility penalty, representing loneliness or reduced social contact, which can be estimated using survey-based measures of well-being [83]. Similarly, cognitive biases, such as misperceptions of infection risk, overconfidence, or time-inconsistent preferences, can be introduced by adjusting belief structures. For example, perceived infection risk could be modeled as a function of actual prevalence, incorporating bias parameters to reflect optimism or pessimism. These features are commonly explored in behavioral economics and psychology [84,85] and can be integrated into FIEM's decision model without disrupting its core structure. These examples highlight how FIEM can use dynamic optimization with additional choice frictions to better approximate the true decision making process and align with other research on the complex determinants of human behavior.

The epidemic model considered here was deliberately simplified to highlight concordance with classic compartmental models, limit the number of parameters, and facilitate interpretation of results. This includes omitting explicit tracking of pre-symptomatic and asymptomatic infectious states, which are particularly important for accurately capturing SARS-CoV-2 transmission dynamics and act to limit individuals' knowledge of their own infectious status. However, FIEM can easily include more complex disease dynamics and health outcomes. For example, we could extend the model to track individuals' knowledge of their infection status (via testing, symptoms, etc.); imperfect reporting, access, and interpretation of data on population-level disease burden; decisions that impact not only contact probabilities but susceptibility to infection, duration of infectiousness, or propensity for severe disease; prosocial behavior in which individuals incur a cost to avoid transmitting disease to others even in the absence of individual risk; and capacity constraints to healthcare resources. Our framework currently classifies individuals into a defined set of strata depending on the combination of their infection state, risk factors, and health decisions, but could readily be extended to an individual-based model, albeit at substantially increased computational cost.

Our model captures the components essential for credible prediction of disease spread and endogenous behavioral responses in a way that is often omitted by other attempts to integrate these two features (e.g., disease spread models with substantial heterogeneity but no explicit individual-level optimization informing behavior, or economic models of human behavior with non-standard epidemiological processes [37,58,61,62,72,73]). However, past work has included other important details that were omitted here but could be integrated into future work. For example, [74] present a rich behavioral model, which captures detailed decisions about time use (i.e., leisure in and outside of home, work in person, and teleworking), the production and consumption of different types of goods (i.e., leisure goods outside of home and consumption within home), and model parameters that are at least partially fitted to match real world data. [73] presents a coupled epi-economics model that formally captures how financial constraints enter the individual's decision problem. Finally, the framework proposed by [62] effectively replicates the economic and epidemiological factors of a specific geography. While FIEM does not currently incorporate these features, future work can extend the framework to capture these valuable model components to improve the specificity of predictions.

The decision to abstain from work to avoid infection was particularly salient during the early phase of COVID-19, when rapid at-home tests, medical-grade face-masks, and vaccines were unavailable. Our model could be extended to consider the additional decisions processes individuals engage in to utilize these interventions, but must include that individuals incur "costs" beyond lost income that may be harder to quantify—such as stigma, social isolation, inconvenience, discomfort, or irrational fears. For other infectious diseases, different decision paradigms arise—to adhere to long-term, nausea-inducing drugs to prevent eventual disease progression or transmission; to lose a potential romantic partner by disclosing an STI, and so on. Our framework allows for extensions in these directions, and we anticipate that the limitation to including them will not be the ability to encode a reasonable model within the FIEM structure, but to identify data appropriate for estimating model parameters. Here we have merely "calibrated" our model—choosing a single reasonable parameter set that roughly recreates a small set of aggregate epidemiological or economic metrics. Future work will present methods for integrating diverse datasets for formal inference of FIEM parameters. Surveys like the COVID-19 Impact Survey [86], which captured income, employment, and health data at the individual level across the U.S. in spring 2020, show that relevant risk group parameters can be estimated rapidly during an outbreak. We hope that case studies using this framework will provide the motivation for behavioral and microeconomic data collection as a

core component of pandemic preparedness activities, so that future disease-behavior models can produce more informed policy recommendations and include uncertainty intervals in all projections.

## Supporting information

**S1 Appendix**. **Supplementary text.** The Supporting Information provides a high-level overview of the general FIEM framework, which integrates individual decision-making with infectious disease dynamics, along with detailed definitions of all variables used in the model. It provides formal descriptions of the decision and infection model components, explains the model parameterization process and data sources, and presents the FIEM algorithm in detail. The document also outlines the policy scenarios evaluated in the study, including labor restrictions, cash transfers, and paid sick leave. Additionally, it lists all simulation parameters, discusses key limitations and assumptions underlying the model, and suggests possible extensions.
(PDF)

## Author contributions

**Conceptualization:** Hongru Du, Matthew V. Zahn, Sara L. Loo, Shaun Truelove, Bryan Patenaude, Lauren M. Gardner, Nicholas Papageorge, Alison L. Hill.

**Data curation:** Hongru Du, Matthew V. Zahn, Sara L. Loo, Alison L. Hill.

**Formal analysis:** Hongru Du, Matthew V. Zahn, Sara L. Loo, Nicholas Papageorge, Alison L. Hill.

**Funding acquisition:** Shaun Truelove, Bryan Patenaude, Lauren M. Gardner, Nicholas Papageorge, Alison L. Hill.

**Investigation:** Hongru Du, Matthew V. Zahn, Sara L. Loo, Shaun Truelove, Bryan Patenaude, Lauren M. Gardner, Nicholas Papageorge, Alison L. Hill.

**Methodology:** Hongru Du, Matthew V. Zahn, Sara L. Loo, Shaun Truelove, Bryan Patenaude, Lauren M. Gardner, Nicholas Papageorge, Alison L. Hill.

**Project administration:** Shaun Truelove, Bryan Patenaude, Lauren M. Gardner, Nicholas Papageorge, Alison L. Hill.

**Resources:** Shaun Truelove, Bryan Patenaude, Lauren M. Gardner, Nicholas Papageorge, Alison L. Hill.

**Software:** Hongru Du.

**Supervision:** Shaun Truelove, Bryan Patenaude, Lauren M. Gardner, Nicholas Papageorge, Alison L. Hill.

**Validation:** Hongru Du, Matthew V. Zahn, Sara L. Loo, Tijs W. Alleman, Nicholas Papageorge, Alison L. Hill.

**Visualization:** Hongru Du, Alison L. Hill.

**Writing – original draft:** Hongru Du, Matthew V. Zahn, Sara L. Loo, Nicholas Papageorge, Alison L. Hill.

**Writing – review & editing:** Hongru Du, Matthew V. Zahn, Sara L. Loo, Tijs W. Alleman, Shaun Truelove, Bryan Patenaude, Lauren M. Gardner, Nicholas Papageorge, Alison L. Hill.

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
