## [Decision Letter · Decision Letter 0]

21 May 2025

PCOMPBIOL-D-25-00647

Improving policy design and epidemic response using integrated models of economic choice and disease dynamics with behavioral feedback

PLOS Computational Biology

Dear Dr. Hill,

Thank you for submitting your manuscript to PLOS Computational Biology. After careful consideration, we feel that it has merit but does not fully meet PLOS Computational Biology's publication criteria as it currently stands. Therefore, we invite you to submit a revised version of the manuscript that addresses the points raised during the review process.

Please submit your revised manuscript within 30 days Jul 21 2025 11:59PM. If you will need more time than this to complete your revisions, please reply to this message or contact the journal office at ploscompbiol@plos.org. Please include the following items when submitting your revised manuscript:

We look forward to receiving your revised manuscript.

Kind regards,

Nicholas Geard

Academic Editor

PLOS Computational Biology

Roger Kouyos

Section Editor

PLOS Computational Biology

**Additional Editor Comments :**

Thank you for your submission, I and the reviewers were impressed by the work presented in your manuscript. While the reviewers were enthusiastic about the quality and potential significance of the research, they have identified a number of minor clarifications and adjustments to enhance the manuscript (including with some of the figures). I don't believe that any of these individually should be onerous; however, there are quite a few, so I encourage you to consider and respond to each carefully.

**Journal Requirements:**

Potential Copyright Issues:

i) Figures 1, and 6B. Please confirm whether you drew the images / clip-art within the figure panels by hand. If you did not draw the images, please provide (a) a link to the source of the images or icons and their license / terms of use; or (b) written permission from the copyright holder to publish the images or icons under our CC BY 4.0 license. Alternatively, you may replace the images with open source alternatives. See these open source resources you may use to replace images / clip-art:

2) If any authors received a salary from any of your funders, please state which authors and which funders..

**Reviewers' comments:**

Reviewer's Responses to Questions

**Comments to the Authors:**

**Please note that one of the reviews is uploaded as an attachment.**

Reviewer #1: Referee report for Improving policy design and epidemic response using integrated models of economic choice and disease dynamics with behavioral feedback.

Overview:

This paper provides an interesting epidemiological-economic model for describing the spread of an infectious disease and a measure of agent economic stress. They employ an individual agent-based decision process to inform risk groups and couple this with a population level model for disease spread, with decisions and disease spread occurring over (potentially) different time steps. As the authors describe, this framework can allow for quite general models to inform future policy decision balancing health and wealth outcomes. The authors provide a nice example of how their framework might be used with decision makers to co-design policy – the biggest barrier to this as it stands is the authors’ model is not trained or validated on real world data. I think this is a well written and presented paper, and if my few comments below are adequately addressed I wholeheartedly support it for publication.

Major Comments:

1. I enjoyed the breadth of discussion in the introduction, but the novelty of the current work should be better identified. I suggest moving the paragraph in “Development of a feedback-informed epidemiological model” to the introduction to help this.

2. The model presented here is a novel proof of concept and has, for example, not been validated with real-world data. This point should be made clearer to the reader from the outset.

3. Figure 1 is hard to follow with a lot going on and not much guidance to assist parsing the information. I suggest reviewing this figure to see if a simpler alternative is possible; alternatively, add to Figure 1 to help with understanding.

4. I am confused by Figure 2 B). This seems to show the average time infectious, which is presumed constant at approximately 7 days in the model description. However, across all scenarios depicted here, it is 8 days or more on average. This suggests some misspecification in the underlying model. Please address this.

5. It would be interesting to see psychological components added to this model to investigate whether they have any potential influence on the agent’s decision-making process. A paragraph on this in the discussion would be nice.

Minor Comments:

1. Please provide references to support the claims in the first paragraph (p.1 ll.45-53).

2. P.4 Section “Decision Scenario and parameterization”. Please provided references and justification for the epidemiological and economic parameter values. For example, why assume R_0=2.6? Why assume a non-vulnerable individual is willing to pay $2000 per day to avoid infection?

3. P.4 ll.184 (and throughout). Please specific the currency being used in the economic model. For example, is this in USD, AUD, NZD?

4. P.6 ll.215 – What pre-defined sensitivity range?

5. In Figure 2, the baseline parameters show an average labor participation of 72%, but table 1 has this as 66.7%. Please explain why these are different.

6. P.15. In the specification of Equation (2) it seems the variable z_{mt} is unnecessary and just adds confusion, as this only appears implicitly in the indicator i_{mt}. I would suggest revising this for clarity.

7. Throughout, the authors have used the utility function as a measure of well-being, but this utility fails to account for many psychological aspects of well-being (e.g. https://doi.org/10.5502/ijw.v13i4.3665). I suggest rephrasing to “financial well-being” or similar as this is what is more accurately captured in their utility function.

Grammar and stylistic comments:

1. P.1 ll.57 spell out Susceptible-Infectious-Recovered (SIR) the first time it appears.

2. P.7 l.226 – some punctuation is missing, i.e. “…labor supply of susceptible individuals infected individuals…”

3. Figure 5: in the caption, the authors say that labor restrictions goes from red to yellow, but the figure shows this as yellow to red.

4. Figure 5: What is the star in E and F?

5. Figure 6 – In the caption when defining U_C, the cost utility should be an upper case C.

6. Figure 7 – It should be a capital B) rather than b).

7. P.15 l.468 there should be a space between “and” and \theta_v.

Reviewer #2: See attached letter to authors.

Reviewer #3: This is a potentially important attempt to develop behavioural-epi models; the authors couple an SIRS model for infection with waning immunity, to a dynamic model for decision to work during a pandemic. Decisions to work are assumed rational, and exponentially discounted, derived from perfect knowledge of individual's own infection status and global prevalence, with risks and benefits of work stratified by socio-economic status, employment status and vulnerability.

**Comments**

It is great to see an attempt to capture heterogeneities in baseline behaviour and behavioural choice during a pandemic. The risk categories within the model are well-motivated in order to enable consideration of a range of potential policy decisions around work during a pandemic. I can see that even this epidemiologically simple (but overall complex) model, could be useful as a tool to communicate trade-offs to policy makers. Regarding the population stratification, do data to parameterise these risk groups exist for most countries? If so that may be something to emphasise as a strength of this approach.

The focus in construction of the utility function is economic costs, have the authors considered modelling the emotional cost of lack of social contact? Would this be possible in this framework?

While the model allows for 8 different categories of agent, with potential for heterogeneous mixing between these strata, the epidemiological model is potentially overly simplistic for a pathogen like SARS-CoV-2: latent and asymptomatic infectious states would generally be considered crucial for describing SARS-CoV-2 transmission, and calibrating using data streams such as reported infection, and indeed realistically capturing **knowledge** about an agent's infectious state. I can see this is partially discussed in in the paragraph beginning line 410, but I'm unsure how you expect this to affect your results? Are the computational costs involved in extending the model prohibitive?

The models for individual behavioural choice are rooted in economic theory and justified via their previous use in this field. I am not an expert in this but I would have liked to see further discussion of the limitations of this approach, including discussion of whether this 'dispassionate rationality' is still considered plausible (see e.g. Epstein & Chalen, 2006)?

There is much discussion of model validation in the supplementary material, but this seems to focus on qualitative trends in model output with parameter changes? Were quantitative unit tests performed to test epidemiological and behavioural aspects of the model? E.g. when disabling waning immunity is the final size as you would expect for a population with one higher risk group?

The authors adopt an agent-based model, which by their nature are extremely flexible, and the authors emphasise this feature as an advantage of their modelling approach while exploring some of the computational limitations of including dynamic feedback in this approach. If the authors felt able, discussion of the computational feasibility of some of the suggested extensions, e.g. spatial heterogeneities, collective decision making, would be very interesting.

While forecasting is deemed beyond the scope of this paper and noted as an area of future work, is it worth discussing the type of data that would allow this model to by verified/falsified, i.e. by expanding on the comments in line 397?

References

------------

Epstein, J., & Chelen, J. (2016). Advancing agent_zero. _Complexity and evolution: toward a new synthesis for economics_, _19_, 299.

**Have the authors made all data and (if applicable) computational code underlying the findings in their manuscript fully available?**

Reviewer #1: Yes

Reviewer #2: **No: **I am not aware of availability of the code; there are no data (sample for estimation) for this project.

Reviewer #3: Yes

PLOS authors have the option to publish the peer review history of their article (what does this mean?). If published, this will include your full peer review and any attached files.

Reviewer #1: No

Reviewer #2: No

Reviewer #3: No

**Figure resubmission:**
---

## [Decision Letter · Decision Letter 1]

7 Sep 2025

PCOMPBIOL-D-25-00647R1

Improving policy design and epidemic response using integrated models of economic choice and disease dynamics with behavioral feedback

PLOS Computational Biology

Dear Dr. Du,

Thank you for submitting your manuscript to PLOS Computational Biology. After careful consideration, we feel that it has merit but does not fully meet PLOS Computational Biology's publication criteria as it currently stands. Therefore, we invite you to submit a revised version of the manuscript that addresses the points raised during the review process.

Please submit your revised manuscript within 30 days Nov 07 2025 11:59PM. If you will need more time than this to complete your revisions, please reply to this message or contact the journal office at ploscompbiol@plos.org. Please include the following items when submitting your revised manuscript:

We look forward to receiving your revised manuscript.

Kind regards,

Roger Dimitri Kouyos

Section Editor

PLOS Computational Biology

**Reviewers' comments:**

Reviewer's Responses to Questions

**Comments to the Authors:**

**Please note that one review is uploaded as an attachment.**

Reviewer #1: I was reviewer 1 on the original submission.

The authors have done a great job at answering all reviewer comments. The only one that is not answered fully is comment 2.31 (Venerable was only corrected in one of the labels in the figure).

Reviewer #2: additional comments in uploaded document

**Have the authors made all data and (if applicable) computational code underlying the findings in their manuscript fully available?**

Reviewer #1: Yes

Reviewer #2: Yes

PLOS authors have the option to publish the peer review history of their article (what does this mean?). If published, this will include your full peer review and any attached files.

Reviewer #1: No

Reviewer #2: No

**Figure resubmission:**
---

## [Editor Report · Decision Letter 2]

23 Sep 2025

Dear Mr Du,

We are pleased to inform you that your manuscript 'Improving policy design and epidemic response using integrated models of economic choice and disease dynamics with behavioral feedback' has been provisionally accepted for publication in PLOS Computational Biology.

Best regards,

Nicholas Geard

Academic Editor

PLOS Computational Biology

Roger Kouyos

Section Editor

PLOS Computational Biology

---

## [Editor Report · Acceptance letter]

PCOMPBIOL-D-25-00647R2

Improving policy design and epidemic response using integrated models of economic choice and disease dynamics with behavioral feedback

Dear Dr Du,

I am pleased to inform you that your manuscript has been formally accepted for publication in PLOS Computational Biology. Your manuscript is now with our production department and you will be notified of the publication date in due course.

With kind regards,

Zsofia Freund
